# The Role of Magnesium in the Pathogenesis of Metabolic Disorders

**DOI:** 10.3390/nu14091714

**Published:** 2022-04-20

**Authors:** Marta Pelczyńska, Małgorzata Moszak, Paweł Bogdański

**Affiliations:** Department of Treatment of Obesity, Metabolic Disorders and Clinical Dietetics, Poznań University of Medical Sciences, 84 Szamarzewskiego Street, 60-569 Poznań, Poland; mmoszak@ump.edu.pl (M.M.); pbogdanski@ump.edu.pl (P.B.)

**Keywords:** magnesium, deficiency, body weight, obesity, hypertension, diabetes, metabolic syndrome, dyslipidemia, inflammation

## Abstract

Magnesium (Mg) is an essential nutrient for maintaining vital physiological functions. It is involved in many fundamental processes, and Mg deficiency is often correlated with negative health outcomes. On the one hand, most western civilizations consume less than the recommended daily allowance of Mg. On the other hand, a growing body of evidence has indicated that chronic hypomagnesemia may be implicated in the pathogenesis of various metabolic disorders such as overweight and obesity, insulin resistance (IR) and type 2 diabetes mellitus (T2DM), hypertension (HTN), changes in lipid metabolism, and low-grade inflammation. High Mg intake with diet and/or supplementation seems to prevent chronic metabolic complications. The protective action of Mg may include limiting the adipose tissue accumulation, improving glucose and insulin metabolism, enhancing endothelium-dependent vasodilation, normalizing lipid profile, and attenuating inflammatory processes. Thus, it currently seems that Mg plays an important role in developing metabolic disorders associated with obesity, although more randomized controlled trials (RCTs) evaluating Mg supplementation strategies are needed. This work represents a review and synthesis of recent data on the role of Mg in the pathogenesis of metabolic disorders.

## 1. Introduction

Magnesium (Mg) is a critical mineral in the human organism involved in regulating many physiological functions. This micronutrient acts as a cofactor or activator in more than 300 enzymatic reactions, participates in RNA and DNA synthesis, protein, lipids and carbohydrate metabolism, stability of cell membranes, bone and calcium (Ca) metabolism, or nervous and immune system functioning [1,2]. To guarantee the correct functioning of the above-mentioned processes, the dietary supply of Mg in food and beverages is required. The recommended daily allowance (RDA) for Mg intake is 320 mg and 420 mg for adult females and males, respectively [3]. In turn, the European Food Safety Authority (EFSA) set the adequate intake (AI) of Mg at 350 mg/day for men and 300 mg/day for women [4]. Unfortunately, epidemiological data show that the daily allowance of Mg is usually unmet in most populations, mostly due to unhealthy dietary patterns, especially the so-called “Western diet” [2,5]. Symptoms of Mg deficiency are often non-specific and may be confounded by low consumption of other nutrients. The clinical diagnosis of Mg deficiency has also become a challenge because its serum concentration does not reflect the total content in the human body [6]. Moreover, Mg deficiency is associated not only with adverse health effects but also with a range of diseases such as cardiovascular diseases (arrhythmia, preeclampsia, heart failure), neurological diseases (headache, seizures, stroke), respiratory diseases (bronchial asthma, chronic obstructive pulmonary disease) and depression [7].

In recent years, special attention has been paid to the participation of Mg in the pathogenesis of metabolic disorders. Nevertheless, the exact mechanism of its action in this area is not fully understood. Some studies indicate that chronic Mg deficiency may be associated with an increased risk of various preclinical and clinical manifestations such as overweight and obesity, insulin resistance (IR) and type 2 diabetes mellitus (T2DM), hypertension (HTN), changes in lipid metabolism, and atherosclerosis, and, as a consequence, with a high risk or cardiovascular diseases (CVD) [8,9,10]. The above-mentioned disorders share a key pathophysiological component attributable to chronic low-grade inflammation (LGI). The protective action of Mg includes attenuating inflammatory processes, improving glucose and insulin metabolism, enhancing endothelium-dependent vasodilation, and normalizing the lipid profile [11]. Thus, it currently seems that Mg plays an important role in the development of metabolic disorders associated with obesity, and for this reason, we undertook this study.

This work represents a review and synthesis of recent data on the role of Mg in the pathogenesis of metabolic disorders.

## 2. The Assessment of Magnesium Status

In the human body, Mg is mostly located in the bones and teeth (about 60%) as well as in the intracellular space (about 40%), i.a., muscles and soft tissues, while <1% is in the blood. In clinical practice, the most common method used to assess Mg status is the evaluation of its concentration in the serum, with the reference range between 0.75 and 0.95 mmol/L (Table 1) [12]. As mentioned above, only 0.8% of this micronutrient is found in the human blood representing 0.3% in the serum and 0.5% in the erythrocytes [2,7]. Thus, it seems that this method poorly reflects the total Mg level in the body. The latter can be influenced by, i.a., dietary pattern, especially Mg intake, albumin levels, or Mg supplementation, which affect the total amount of Mg absorbed and excreted through the kidneys. In subclinical Mg deficiency, blood levels may not deviate from the norm. On the contrary, a serious Mg absence in the tissues and bones may be present. It is estimated that subjects with Mg concentrations in the lower range of the norm (<0.8 mmol/L) are likely to have a deficiency of this micronutrient. Therefore, they should be tested with alternative methods used to evaluate Mg levels as its serum concentration does not reflect the total Mg content in the tissues and organs. Those situations may lead to the underestimation of Mg deficiency both in healthy and diseased individuals [13,14].

The diagnosis of Mg deficiency is challenging. The more accurate method to evaluate Mg concentration in the human body is the red blood cell (RBC) Mg levels (ranges between 4.2 and 6.8 mg/dL) (Table 1) [15]. In Mg deficiency, this micronutrient is pulled out from the RBC cells to maintain blood Mg levels within the normal range. It is recommended not to supplement any minerals within one week before the analysis [6,15]. It is worth mentioning that some studies showed that this method is useful only in long-term Mg repleted or depleted dietary patterns (over three months) as well as questioning the validity of this method [16,17].

From other methods used to assess Mg status, the urine Mg levels are poorly correlated with the amount of this micronutrient in the human body because of the wide daily fluctuation of reabsorption and secretion of Mg through the kidneys [6]. On the other hand, a meta-analysis of randomized controlled trials demonstrated significant dose and time responses of circulating Mg concentration and 24-h urine Mg excretion to its oral supplementation [18]. The other methods remain intravenous or oral Mg loading tests followed by a 24-h urine collection with the assessment of Mg excretion. Some studies showed that the load retention over 27% (with the range in healthy individuals around 2–8%) indicates an Mg deficiency [19,20]. On the contrary, some analyses criticize this technique because of clinical unsuitability, high costs, and doubtful standardization [21,22]. Sometimes, non-commercial techniques are used to assess Mg status such as Non-invasive Intracellular Mineral-Electrolyte Analysis (EXA) or a hair mineral analysis test as well as isotopic Mg labels [16]. Finally, it is suggested that ionized serum Mg concentration [23] as well as serum Mg/Ca ratio (with the optimal range at 0.4) are also practical indicators of Mg status and its turnover in the human body (Table 1) [24].

**Table 1 nutrients-14-01714-t001:** Key elements in magnesium deficiency.

Serum Mg Concentration	0.75–0.95 mmol/L [12]
Methods of Mg status evaluation in human body	Mg serum concentration RBC Mg levels Mg urine levels Intravenous or oral Mg loading tests Non-invasive Intracellular Mineral-Electrolyte Analysis Hair mineral analysis test Using isotopic Mg labels Ionized serum Mg concentration Serum Mg/Ca ratio [6,15,16]
Mg recommended daily allowance	Children aged 1–3: 80 mg/day Children aged 4–8: 130 mg/day Adolescents aged 9–13: 240 mg/day Girls aged 13–18: 360 mg/day Boys aged 13–18: 410 mg/day Adult women: 320 mg/day Adult men: 420 mg/day [3]
Mg deficiency symptoms	Early signs of Mg deficiency: fatigue, weakness, loss of appetite, nausea or vomiting Advanced Mg deficiency: tremor, agitation and muscle fasciculation, cramps, seizures, cardiac arrhythmia, ventricular tachycardia, personality changes or depression [25]
Mg food sources	Almonds, bananas, black beans, green vegetables (spinach, broccoli), nuts, oatmeal, seeds, brown rice, unprocessed cereals, soybeans, sweet corn, tofu, and dark chocolate [15,26]

Abbreviation: Ca, calcium; Mg, magnesium; RBC, red blood cells.

It seems that, even today, no single method of Mg deficiency evaluation is considered satisfactory. The difficulties of accessing total body Mg concentration concerns its main two compartments, namely bone and muscle, while in blood it is present only in very small amounts (less than 1%). It seems that serum Mg concentration still remains a useful measurement for the diagnosis of its deficiency. It is worth considering extending the diagnostics of Mg deficiency by loading tests, which also appear to be an accurate laboratory test to indicate the Mg status in the human body [16]. Nevertheless, to comprehensively assess this micronutrient, both laboratory techniques and clinical evaluation of the patient’s condition are required.

## 3. Magnesium Deficiency

Epidemiological studies conducted in Europe and North America show that Mg consumption is usually lower than recommended amounts (about 30–50%), especially as a result of inappropriate nutrition, such as so-called Western-type dietary patterns [27,28]. There are also other causes such as agronomic and environmental factors, affecting the Mg content and availability in the soil and, consequently, in the food chain [29]. Often, together with an inadequate intake of Mg, a decrease in intestinal absorption or increased excretion of this micronutrient causes hypomagnesemia [2]. Data indicate that around 10%–30% of the population may have a subclinical Mg deficiency, based on its serum levels (<0.80 mmol/L) [30]. The results obtained from the National Health and Nutrition Survey showed low serum Mg concentrations in more than 30% of the Mexican adult population [31]. The coexistence of chronic illnesses often exacerbates Mg deficiency. In a systematic review and meta-analysis of experimental studies, authors showed that subjects with prediabetes had significantly lower serum Mg concentration than healthy controls (about 0.07 mmol/L) [32]. Another study estimated that the prevalence of hypomagnesemia at intensive care units may be around 59.3% and is associated with a slightly higher mortality rate in elderly patients [33]. It is worth mentioning that Mg deficiency can also be present despite the normal serum concentration of this micronutrient. The study conducted by Nielsen et al. [5] suggested that individuals with serum Mg concentrations between >0.75 mmol/L (1.82 mg/L) and <0.85 mmol/L (2.06 mg/dL) may have an Mg deficiency, especially if they have a poor Mg intake from their diet (<250 mg/day) and urinary excretion of <80 mg/day.

As mentioned before, the RDA for Mg intake of various population groups was established in 1997 by the Institute of Medicine. The recommendation of Mg intake depends on age, and for small children aged 1–3 and 4–8, it is 80 mg/day and 130 mg/day, respectively. For adolescents (aged 9–13), it takes 240 mg/day, and 360 and 410 mg/day for girls and boys, respectively, aged 13–18, with final levels of 320 and 420 mg/day for adult women and men, respectively (Table 1) [3]. The intestine, bone, and kidneys are the main organs responsible for maintaining Mg homeostasis. This micronutrient is mainly absorbed in the small intestine, stored in bones, and excreted by the kidneys and feces. Although the majority of Mg is absorbed in the small intestine by a passive paracellular mechanism, a minor amount is transported via the transcellular transporters transient receptor potential channel melastatin members TRPM 6 and TRPM 7, which still remain an important transport mechanism of this micronutrient [34,35]. Only about 24–76% of the total Mg supply is absorbed in the intestines, and the rest is eliminated in the feces [26]. Additionally, the kidneys play an important role in the regulation of Mg status in the human body. The homeostasis of Mg is as follows: daily the intestines absorb ≈120 mg and secrete 20 mg of Mg, resulting in a net absorption of 100 mg of Mg, while in the kidney daily ≈2400 mg of Mg is filtered by the glomerulus, of which 2300 mg (≈95%) is reabsorbed along the kidney tubule. That also results in a net excretion of 100 mg (≈3–5%) of Mg, and this matches the intestinal absorption. Mg is stored in bones (up to 50–60%), muscles (up to 25–30%), and other tissues (up to 20–25%) (Figure 1) [36]. It is worth considering that intestinal absorption is not directly proportional to Mg intake, but it depends on its status. The lower the levels of this micronutrient, the more it is absorbed in the intestines; thus, its absorption is high when intake is low and vice versa [2]. Mg homeostasis is also influenced by different hormones, especially 1,25-dihydroxyvitamin D (1,25(OH)_2_D) and parathormone (PTH); thus, their inappropriate levels may correlate with abnormalities in Mg concentration [34,37].

It has been observed that the active form of vitamin D, i.e., 1,25(OH)_2_D, can stimulate the absorption of Mg in the intestine. In turn, Mg acts as a cofactor and regulates several steps in vitamin D metabolism. Mg is required for the binding of vitamin D to its transport protein as well as hepatic and renal hydroxylation. Mg deficiency leads to 1,25(OH)_2_D reduction and an impaired PTH response [37,38]. The cause–effect relationship between serum PTH level and Mg concentration seems to be complex. The PTH secretion is physiologically controlled by the serum Ca levels, but Mg can exert similar effects as well. Low Mg levels stimulate PTH secretion, while very low serum Mg concentrations induce a paradoxical block. The latter leads to clinically relevant hypocalcemia in severely Mg deficient patients. On the other hand, PTH regulates Mg homeostasis by modulating its renal reabsorption, absorption in the gut, and release from the bone [39].

The clinical signs of hypomagnesemia are often non-specific (Table 1). Early signs of Mg deficiency may be fatigue, weakness, loss of appetite, nausea, or vomiting. As Mg deficiency worsens, the next symptoms may include tremors, agitation and muscle fasciculation, cramps, seizures, cardiac arrhythmia, ventricular tachycardia, personality changes, or depression [25]. In different clinical situations, Mg deficiency symptoms may be latent. Its high incidence is usually observed in intensive care units [40]. In addition, clinical symptoms of hypomagnesemia are often correlated with a rapid decrease in Mg levels compared to a gradual change. Hypomagnesemia is frequently accompanied by other electrolyte abnormalities, especially hypokalemia and hypocalcemia. Thus, the diagnosis of Mg deficiency should be supported by laboratory determinations of other macroelements, i.a., calcium, potassium, or phosphorus [34]. Conditions leading to hypomagnesemia, as well as insufficient dietary supply, include malabsorption, endocrine causes, renal disease, redistribution and intracellular shift of Mg, medication use, or other factors such as chronic alcoholism or stress (Table 2) [2,26,41].

It is important to consume an adequate amount of Mg through diet. The good sources of this micronutrient include green vegetables, nuts, seeds, unprocessed cereals, and dark chocolate (Table 1). A lower concentration of Mg is present in fruits, meat, fish, and dairy products. Drinking water supplies about 10% of daily Mg intake [26]. It is worth adding that dietary factors such as lactose, fructose, or glucose can enhance Mg absorption, while a high intake of zinc, fiber, free fatty acids, oxalate, or phytate can cause its decrease [30].

## 4. An Association of Magnesium with Excessive Body Weight

Obesity and its comorbidities have become a relevant medical problem worldwide, and data from epidemiological studies show that their prevalence is still growing. Obesity has increased worldwide in the last 50 years, reaching a pandemic level. According to the World Health Organization (WHO), more than 1.9 billion people worldwide are overweight, and over 650 million people are obese. The prevalence of obesity is higher in women (15% of the population) than in men (11% of the population) [42]. Obesity can be diagnosed with a body mass index (BMI) over 30 kg/m^2^, waist circumference (WC) over 80 cm for Caucasian women and 94 cm for men [43], or a high content of adipose tissue content usually measured with the use of a bioelectrical impedance analysis (BIA) and less frequently with a Dual Energy X-ray Absorptiometry (DXA) method (dependent on sex and age) [44].

Changes in Mg metabolism have been observed in obese patients, leading to the reduction in the serum, plasma, and erythrocytes concentration of this micronutrient [45]. A recent meta-analysis showed that overweight and obese women (with BMI ≥ 25 kg/m^2^) with polycystic ovary syndrome had lower Mg concentrations (weighted mean differences (WMD): −0.07, 95% CI: −0.14, −0.01 mmol/L; *p* = 0.02) than normal-weight controls ((WMD: −0.11, 95% CI: −0.25, 0.04) mmol/L; *p* = 0.14) [46]. Another population-based cross-sectional study involving 130 healthy adults found a significant negative correlation between body weight (correlation coefficient-r, r = −0.30, *p* = 0.003) and WC (r = −0.21, *p* = 0.03) and total Mg serum levels [47]. It is indicated, that not only adult obese subjects can be predisposed to Mg deficiency but children as well. The analysis conducted by Hassan et al. [48] evaluated a significantly lower Mg concentration in overweight and obese children (2.08 ± 0.211 mg/dL) as compared to the normal-weight participants (2.55 ± 0.155 mg/dL, *p* < 0.001). Moreover, the authors showed a strong inverse correlation between Mg serum levels and BMI. Obesity often results from unhealthy diets, rich in calories and poor in essential nutrients. Data from the National Health and Nutrition Examination Survey (NHANES) showed that Mg intakes were negatively correlated with BMI and WC, after adjusting for age and gender [49]. In another longitudinal study it was suggested that Mg intake was inversely associated with the incidence of obesity [50] (Table 3).

The possible mechanism underlying Mg deficiency and the occurrence of excessive body weight may include the decreased absorption or increased excretion of this micronutrient as a result of an inappropriate diet with insufficient content of fiber, whole grains, or green leafy vegetables, i.e., good sources of Mg [47,48]. Obesity results from unhealthy diets, high in calories but poor in essential nutrients. As a consequence, obese subjects are often Mg deficient [10]. A poor Mg intake impairs intestinal absorption and promotes a pro-inflammatory response in overweight and obese individuals. Intestinal inflammation, in turn, impairs micronutrient absorption [76,77]. Moreover, a higher Ca or fatty acids intake with diet can interfere with Mg absorption in the intestines [48]. It is also hypothesized that this micronutrient may have an anti-obesity effect because of its ability to form soaps with fatty acids in the intestine and thus reduce the absorption of fatty acids from the diet [78,79]. The next issue is the effect of hypoxia as a result of adipose tissue enlargement. Although the clinical studies evaluating the effect of Mg supplementation on the accumulation of adipose tissue are still missing, the results from animals models indicate that it protects from this phenomenon [80]. Hypoxia may induce the synthesis of adipocytokines and pro-inflammatory compounds such as high-sensitivity C-reactive protein (hs-CRP), plasminogen activator inhibitor-1 (PAI-1), tumor necrosis factor-α (TNF-α), or interleukin-6 (IL-6) [81], while human studies indicate an inverse relationship between the Mg concentration and LGI [82,83]. The latter is often accompanied by obesity, although not in every person with an excessive body weight is an increase in pro-inflammatory indicators observed. Thus, some nutritional factors such as a low Mg supply may contribute to this disorder.

Chronic Mg deficiency causes the reduction in extracellular Mg levels and an increase in intracellular Ca concentration as well as the priming of phagocytic cells, which results in the release of inflammatory cytokines [84]. Some studies that use experimental animal models, support the findings that subclinical Mg deficiency can promote chronic inflammatory reactions in humans as a result of the cellular calcium entry and abnormalities of its signaling contributing to the releasing of inflammatory molecules as neuropeptides, cytokines, prostaglandins, and leukotrienes [85,86]. The mechanism by which Mg regulates Ca entry to the cell occurs in the N-methyl-D-aspartate (NMDA) receptor. The reduction in extracellular Mg concentration lowers the threshold of excitatory amino acids, such as glutamate, needed to activate the above-mentioned receptor. The activation of the NMDA receptor leads to the influx of Ca into the cell. In turn, the blockade of the NMDA receptor decreases the pro-inflammatory cytokines in the plasma (prostaglandin E2) and lowers the cardiac inflammation molecules in the heart, which are promoted by Mg deficiency [87]. The appropriate vitamin D level is also important because it correlates with Mg status in obese individuals. Obesity is often correlated with vitamin D deficiency, and Mg is essential for the synthesis and activation of this vitamin [88]. Thanks to pleiotropic properties, vitamin D decreases the risk of metabolic-related disorders in obese subjects [89]. In the study conducted by Stokic E. et al. [90], it was shown that chronic vitamin D deficiency together with low Mg levels predispose non-diabetic obese individuals to an increased risk of cardiometabolic disorders, while maintaining an optimal Mg level improves the protective action of this vitamin (Figure 2).

As mentioned above, data from human studies indicate that a high Mg intake with diet [53], as well as its serum concentration [47], shows an inverse association with markers of adiposity, such as BMI and WC. Moreover, supplementation of this micronutrient seems to protect against obesity and its comorbidities. In a recent meta-analysis of RCTs conducted by Askari M. et al. [91] it was shown that Mg supplementation resulted in a slight reduction in BMI, especially in those with Mg deficiency, insulin resistance-related disorders, and obesity at baseline. In another systematic review and dose-response meta-analysis of clinical trials [92] it was observed that there were no significant changes in anthropometric indicators after Mg supplementation in the overall analysis; nevertheless, in the subgroup, Mg supplementation decreased WC in obese individuals. On the other hand, a meta-analysis conducted by Asbaghi et al. [93] showed that Mg supplementation did not significantly change body weight, BMI, or WC in obese patients with T2DM compared to the control group (Table 4). It is also worth remembering that some genetic variations (*ARL15* rs35929) may modify the association of Mg status with fat mass in a general population [94]. To sum up, in a number of obese patients, Mg supplementation may protect against weight gain, although more studies in this area are required.

## 5. Magnesium and Hypertension

Hypertension (HTN) is considered the strongest independent and modifiable risk factor for coronary diseases such as heart failure, stroke, myocardial infarction, or chronic kidney disease [109]. According to WHO, 1.28 billion adults aged 30–79 years worldwide have HTN, and its prevalence is still increasing. It is estimated that about 46% of the adult population with HTN is unaware of the condition [110]. Therefore, HTN is an emerging health problem, and new strategies are needed to fight its pandemic. Data from observational studies have revealed an inverse correlation between Mg serum concentration and intake with diet and HTN [111,112].

A meta-analysis of a prospective cohort of 11 studies, including five reports’ results on coronary heart diseases (n-38,808 individuals with an average 10.5-year follow-up) and three reports on HTN (14,876 participants with a 6.7-year follow-up), showed that comparing the highest to the lowest category of circulating Mg concentration, the pooled relative risks (RR) were 0.86 (95% CI: 0.74, 0.996) and 0.91 (95% CI: 0.80, 1.02) for the incidence of coronary heart diseases and HTN, respectively. Moreover, every 0.1 mmol/L increment in circulating serum Mg levels was associated with a 4% reduction in HTN incidence (RR: 0.96; 95% CI: 0.94, 0.99) [113]. In another systematic review and meta-analysis of prospective cohort studies covering nine articles (six on dietary Mg intake, two on serum Mg concentration and one mixed), including 20,119 cases of HTN, an inverse association between dietary Mg intake and the risk of hypertension comparing the highest intake group with the lowest was found (RR: 0.92; 95% CI: 0.86, 0.98). Moreover, a 100 mg/day increment in Mg intake was associated with a 5% reduction in the risk of HTN (RR: 0.95; 95% CI: 0.90, 1.00). Nevertheless, the association of serum Mg concentration with the risk of HTN was marginally significant (RR: 0.91; 95% CI: 0.80, 1.02) (Table 4) [111]. In the SUN Project, it was observed that dietary Mg supply < 200 mg/day was independently associated with a higher risk of developing high blood pressure, especially in overweight/obese participants (Table 3) [54]. On the other hand, not all studies support the hypothesis that a low serum Mg level is a risk factor for developing HTN or CVD [114].

A low dietary Mg supply resulting in hypomagnesemia might be a contributing factor in the pathophysiology of HTN. Mg deficiency may promote cell dysfunction and increase the potential risk of thrombosis and atherosclerosis. Mg diminishes vascular tone and resistance by releasing the nitric oxide (NO) from the coronary endothelium as well as antagonizing the effect of vasoconstrictor molecules such as calcium, bradykinin, angiotensin II, or serotonin [36]. The experimental studies have proved the relationship between dietary Mg intake or its supplementation and biomarkers of endothelial function. In one cross-sectional study, Mg intake was inversely associated with plasma concentrations of E-selectin (*p* = 0.001) and soluble intercellular adhesion molecule 1 (sICAM-1) (*p* = 0.03) [73]. In another analysis, independent of known risk factors for metabolic outcomes, an inverse relationship was associated between Mg intake and plasma concentrations of soluble vascular cell adhesion molecule 1 (sVCAM-1) and E-selectin [57]. Moreover, in a meta-analysis of randomized controlled trials, including seven studies (n-306), it was evaluated that Mg supplementation significantly increased flow-mediated dilation—FMD (WMD: 2.97; 95% CI: 0.23, 5.70, *p* = 0.033) in the general population [115].

As mentioned before, Mg acts as a natural calcium channel blocker. In physiological conditions, extracellular Mg concentration inhibits Ca entry into the cells. Calcium influx across the external cellular membrane in smooth muscle cells and cardiomyocytes is in turn necessary for the contraction of these cells and the regulation of vascular tone. Moreover, intracellular Ca and Mg levels are regulated by reversible binding to specific calcium-binding proteins, while their flux across the external membrane is controlled by a calcium–magnesium–ATPase pump as well as Ca channels. It was investigated that in hypertensive patients, an increased concentrations of Ca and decreased levels of Mg in cell membranes are observed [2,116,117]. Thus, owing to the modulation of a variety of signaling pathways, it is possible that the altered Mg concentrations may participate in smooth muscle cells’ contraction and relaxation, whereby they may modify blood pressure values [14]. It is worth adding that Mg levels can also modulate vascular tone and blood pressure through anti-oxidant and anti-inflammatory effects. The adequate Mg levels probably reduce reactive oxygen species (ROS) formation and limit the vasoconstriction effect. The potential mechanism may involve the modulation of the expression of solute carrier family 41 member (1SLC41A1), the main exchange mechanism responsible for Mg efflux in mammalian cells [118]. It has been reported that gene upregulation of SLC41A1 may be accompanied by Mg deficiency [119]. The latter can activate Mg transporters such as TRPM7 (for Mg influx) and SLC41A1 (for Mg efflux) to induce Mg effluence from cells to the serum to increase Mg levels. This mechanism may affect Mg/MgATP-dependent cellular signaling and its functions. A decreased extracellular Mg concentration can result in Mg releasing from the mitochondria through solute carrier family 41 A3 (SLC41A3, for Mg efflux from mitochondria). Decreased mitochondrial Mg levels could also affect Mg/MgATP-associated mitochondrial signaling and functions, which may explain the mitochondrial ROS overproduction [120,121,122,123] (Figure 2).

Mg supplementation as a therapy to support hypertension remains a subject of discussion, although, as mentioned above, many epidemiological studies confirm the protective action of Mg due to hypertension. In a meta-analysis of randomized, double-blind, placebo-controlled trials in normotensive and hypertensive adults, it was found that Mg significantly reduces systolic blood pressure (SBP) and diastolic blood pressure (DBP) along with increases in serum Mg concentration (Table 4) [96]. In another meta-analysis of RCTs, already mentioned in this manuscript, Asbaghi et al. [93] showed that although Mg supplementation did not affect body weight, it significantly reduced the SBP and DBP in T2DM patients. One more meta-analysis enrolled individuals with insulin resistance, prediabetes, or with non-communicable chronic diseases (NCDs) and found that Mg supplementation resulted in a mean reduction in SBP of 4.18 mmHg and 2.27 mmHg in DBP [95]. Although there is a lack of meta-analysis denying the effect of Mg supplementation on HTN therapy, some analyses found a minimal impact of this micronutrient on blood pressure values (Table 4) [97].

## 6. Magnesium and Diabetes and Metabolic Syndrome

Diabetes mellitus (DM) is a serious chronic disease that occurs at elevated fasting blood glucose (FBG) levels and/or IR. Diabetes is an important public health problem; one of the priority NCDs targeted for research action. Mg deficiency is very common in diabetes mellitus patients (both types) and their occurrence differs from 13.5–47.7% [124]. There is strong evidence linking the association between the metabolic defects in DM and abnormal Mg concentration [125].

Results from the Canadian Health Measures Survey cycle 3 (2012–2013), with subjects aged 3–79 years (n, 5561), showed that serum Mg concentrations in individuals with diabetes was lower (from 0.04 to 0.07 mmol/L) compared to healthy participants. Serum Mg levels were also negatively associated with BMI and diabetes components such as fasting blood glucose and insulin levels, and glycated hemoglobin (HbA1c) as well as the Homeostatic Model Assessment for Insulin Resistance (HOMA-IR) [126]. In a population-based cohort study of 8555 participants (mean age 64.7 years; median follow-up 5.7 years) with normal FBG levels (mean ± SD: 5.46 ± 0.58 mmol/L) at baseline, a 0.1 mmol/L decrease in serum Mg levels was associated with an increase in diabetes risk (hazard ratio, HR: 1.18, 95% CI: 1.04, 1.33). Moreover, an association between serum Mg levels and prediabetes risk was found (HR 1.12, 95% CI: 1.01, 1.25) [127]. As well as the Mg level, the intake of this micronutrient also seems to correlate inversely with the risk of diabetes. A meta-regression analysis of 25 prospective cohort studies (n-637,922) including 26,828 subjects with DM2 showed that after adjusting for age and BMI, the risk of T2DM incidence was smaller by 8–13% for a 100 mg of Mg increment in intake per day [128]. The recent meta-analysis, from the year 2020, also found an inverse association between the risk of T2DM and Mg intake (by 22%, RR: 0.78, 95% CI: 0.75, 0.81) [129]. Other meta-analysis reviews have also confirmed this report [130,131]. The above-mentioned finding suggests that the appropriate consumption of Mg-rich food products such as whole grains, beans, nuts, and green leafy vegetables may reduce the risk of type 2 diabetes.

Diabetes and other metabolic disorders such as excessive body weight, HTN, and dyslipidemia often coexist as metabolic syndrome (MetS). The latter often correlates with Mg deficiency and poor Mg intake. The results of the clinical research also support these data. In a systematic review and meta-analysis conducted by Sarrafzadegan N. et al. [132], it was evaluated that a higher Mg intake is associated with a lower risk of MetS (nine articles, n, 31,876, odds ratio, OR = 0.73; 95% CI: 0.62, 0.86; *p* < 0.001), while serum Mg concentration showed a significant but heterogeneous association with metabolic syndrome occurrence (eight articles, n, 3487, mean difference, MD: −0.19; 95% CI: −0.36, 0.03; *p* = 0.023). Additionally, a dose-response meta-analysis and meta-regression based on eight cross-sectional studies and two prospective cohort studies showed that the pooled relative risks of MetS per 150 mg/day increment of Mg intake were 0.88 (95% CI: 0.84, 0.93; I2 = 36.3%). The meta-regression model indicated a linear, inverse relationship between Mg supply and metabolic syndrome occurrence [133].

Poor Mg intake and/or augmented Mg urinary loss seem to be important causes of Mg deficiency in T2DM patients, while the absorption and retention of this micronutrient is usually maintained in the above-mentioned population [134]. The exact mechanism explaining the role of Mg in the pathogenesis of diabetes is not fully understood, although there are some pathways linking Mg deficiency and disturbances in glucose and insulin metabolism. In addition, the relationship between Mg and insulin is bipartite. On the one hand, insulin regulates Mg homeostasis, but on the other, Mg is a major factor determining insulin and glucose actions. Mg plays a crucial role in carbohydrate metabolism, mainly through its impact on tyrosine kinase (TK) activity of the insulin receptor and regulations of peripheral insulin sensitivity [2,135]. The major pathophysiologic defect involved in the development of T2DM and IR is an impairment in intracellular signaling [136]. The insulin receptor is an integral membrane glycoprotein, formed by two alpha and two beta subunits. In the insulin signaling process, insulin binds to the alpha subunit, which activates the TK in the beta subunit, also autophosphorylation [137]. Mg, in turn, acts as a cofactor by regulating the tyrosine kinase activity of the beta subunit. Insulin actions are carried out by the activation of one of the two main signaling pathways such as: insulin receptor substrate (IRS)/phosphatidylinositol 3-kinase (PI3K)/protein kinase B (Akt). This pathway plays an important role in insulin signaling due to its activation that leads to the phosphorylation of several key substrates, resulting in regulation, i.a., glucose transport or glycogen synthesis. The disturbance in the PI3K/Akt kinase pathway may also lead to a reduction in insulin trafficking and glucose uptake in target tissues stimulated by glucose transporter type 4 (GLUT4) [138]. Furthermore, Mg has also been shown to increase GLUT4 gene expression [139]. The next thing is that in the liver, Mg regulates gluconeogenic enzymes (including glucose-6-phosphatase), while in the adipose tissue, it shows an anti-inflammatory effect. In turn, under low Mg conditions, pro-inflammatory molecules, such as IL-1 and TNF-α, may attenuate IRS-1 and GLUT-4 activation, respectively [136,140]. It is worth mentioning that carbohydrate metabolism is also regulated due to antagonistic actions by Ca and Mg. Extracellular Mg concentration acts as a Ca antagonist and inhibits its influx, necessary for insulin secretion. A low extracellular Mg concentration results in an increased Ca influx and high intracellular concentration of this micronutrient, which stimulates insulin secretion by beta cells; thus, insulinemia seems to be inversely correlated with magnesemia [140,141,142]. To conclude, Mg deficiency has a significant impact on insulin secretion and may contribute to pancreatic beta-cell dysfunction (Figure 2).

It seems that not only Mg serum concentration and its intake with diet can modify the risk of diabetes and metabolic disorders but also the supplementation of this micronutrient may affect the glucose and insulin levels. In databases, a few meta-analyses assessing the effect of Mg supplementation on T2DM can be found. In one of them, supplementation of this mineral provided a significant improvement not only in FBG but also in other metabolic syndrome components such as high-density lipoprotein (HDL), low-density lipoprotein (LDL), plasma triglycerides (TG), and SBP [99]. In another systematic review and meta-analysis of double-blind RCTs, the authors evaluated that Mg supplementation appears to have a beneficial influence on glucose parameters in people with DM2 and insulin sensitivity parameters in people at high risk of diabetes [100]. Those same authors confirmed the above-mentioned results in a very recent meta-analysis, concluding that compared with a placebo, Mg supplementation reduces fasting plasma glucose in people with diabetes and improves insulin sensitivity markers in those at high risk of diabetes (Table 4) [98]. To summarize, hypomagnesemia is often seen in type 2 DM patients as well as being inversely correlated with intensification of the diabetes. Supplementation of Mg seems to prevent chronic metabolic complications related to diabetes mellitus, although more RCTs evaluating i.a. Mg supplementation strategies are needed.

## 7. Magnesium and Dyslipidemia

Dyslipidemia, one of the factors involved in MetS, is a significant public health problem worldwide, leading to the development of atherosclerosis and CVD. Dyslipidemia is defined as increased levels of total cholesterol (TC) and/or TG that may be accompanied by decreased high-density lipoprotein levels. In dyslipidemia, elevated low-density lipoprotein levels (LDL) and excessive hepatic very-low-density lipoprotein (VLDL) production can also be observed. High TC, especially elevated LDL, plays an important role in the pathogenesis of plaque formation, vascular endothelium dysfunction, and promotion of inflammation, which affects atherosclerotic initiation [143,144]. The influence of LDL on progression to atherosclerosis is dependent on the endothelial phenotype (regulation of transcription factor Krüppel-like Factor 2 (KLF2) expression) [145] and serum LDL levels (dose-dependent relation in LDL above 20–40 mg/dL) [146].

The development of lipid profile abnormalities is conditioned by several factors, including genetic background, a Western-style diet high in energy, simple carbohydrates, and saturated fat, excessive alcohol use, overweight/obesity, insulin resistance, and diseases such as nephrotic syndrome [147]. The previous studies also suggested that Mg level may be associated with lipid abnormalities [148,149,150,151,152,153,154,155,156].

The strong relationship between hypomagnesemia and dyslipidemia (OR: 2.8, 95% CI: 1.3, 2.9) was noticed in a cross-sectional study by Guerrero-Romero et al. [148] in which patients with MetS (n, 192) were compared with healthy subjects (n, 384). This study showed that 65.6% of MetS patients had low serum Mg levels, whereas, in the healthy group, Mg deficiency was observed in only 4.9%. The same relationship of hypercholesterolemia in individuals with lower Mg concentration was proved earlier by Bersohn and Oelofse [149]. In addition, the systematic review (nine articles) and meta-analysis (two cross-sectional studies) conducted by Rodrigues et al. indicated that in type 1 diabetes mellitus patients, reduced serum Mg was associated with higher TG, TC, and LDL levels, and lower HDL levels [150]. The relationship between serum Mg concentration and lipid metabolism indicators was also demonstrated in other studies involving different study groups (e.g., children and adults, various diseases) [151,152,153,154].

Conversely, the cross-sectional study aimed at analyzing how serum concentrations of six mineral elements (including Mg) affect the risk of dyslipidemia in Chinese adults (n, 1466), showed no significant differences in serum Mg concentration between dyslipidemia vs. normal-lipidemia subjects. In truth, generally, it was noted that patients with higher than normal TC, TG, HDL, and LDL levels had lower Mg status (*p* = 0.002), but this relation was not significant in an adjusted for gender, BMI, age, glucose, and overall mineral elements model. The lack of correlation between the Mg status in the body and the lipid concentration was also shown in studies by Khatami et al. [155].

In Barragán et al.’s study [156] aimed at analyzing the relation between plasma Mg concentration and cardiovascular risk in adults from a general Mediterranean population (n, 492), interesting conclusions were provided. The prevalence of hypomagnesemia in this population (<0.70 mmol/L) was 18.6%, but there was no significant association between hypercholesterolemia and hypomagnesemia (OR: 1.38, 95% CI: 0.81, 2.35, *p* = 0.233). Moreover, plasma Mg concentrations were higher in hypercholesteremic subjects vs. non-hypercholesteremic (0.79 ± 0.09 mmol/L vs. 0.76 ± 0.07 mmol/L; *p* = 0.001), and hypercholesterolemia was diagnosed more prevalently in patients in the fourth quartile of Mg concentrations (OR: 3.12; 95% CI: 1.66, 5.85, *p* < 0.001). In the study, significant correlations between Mg level and TC (*p* = 0.01) and LDL (*p* = 0.002) but not with fasting TG (*p* = 0.959) and HDL (*p* = 0.933) were also noticed.

It is worth emphasizing that, despite many observational studies with different specifications, it is difficult to conclude unequivocally about the relationship between the blood Mg concentration and the occurrence of lipid profile disturbances.

One of the factors predisposing to the development of dyslipidemia is unhealthy dietary habits, especially consuming a high-calorie, high-fat, and high-sugar diet, deficient in vegetables, fruits, and whole grains [157]. The last three mentioned groups of food products are good dietary sources of Mg. Therefore, it can be assumed that the amount of Mg provided with the diet may be a risk factor for lipid disorders.

In a cross-sectional study including 2504 Tehran adult participants, Mirmiran et al. [67] noticed that higher dietary Mg was inversely correlated with TG level, but there was no association with HDL (Table 3). The relation between Mg intake and TG plasma concentration has also been proved in a retrospective analysis involving a large population of 12,284 individuals from NHANES. In this study, in the subgroup of females, the dietary Mg content was positively correlated with HDL concentration but negatively with the TC/HDL-C ratio [64], which provided evidence that nutrient–lipids associations may be gender specific. The inverse relationship between dietary Mg intake and serum TC levels was observed in the European Prospective Investigation into Cancer (EPIC), Norfolk study [65]. In RCTs that evaluated the effect of an Mg-rich diet vs. a diet with usual Mg content on protection against cardiovascular deaths among high-risk individuals, the Mg–lipids relation was observed. The authors stated that despite the lipid reducing effect of a high-fiber and low-cholesterol diet, dietary Mg intake may contribute to the improvement in lipid parameters as well [70] (Table 3). It is worth noting that the relationship observed in this study concerns a diet with an increased proportion of Mg intake, so it should be compared with studies on Mg supplementation. A large body of studies have also demonstrated the preventive effect of increased dietary Mg on dyslipidemia [66,68,69,70,158] (Table 3).

The potential benefits of dietary Mg supply on dyslipidemia management may be linked to the fact that Mg participates in the modulation of lipoprotein lipase (LPL), desaturase (DS), HMG-CoA reductase, and lecithin-cholesterol acyl transferase (LCAT) activity [103]. Previous studies elucidated that hypomagnesemia suppresses the activity of LPL, DS, and LCAT and Mg supplementation may positively modulate their expression [159,160]. Conversely, the β-hydroxy β-methylglutaryl-CoA (HMG-CoA) reductase activity is elevated in Mg deficiency [161]. Impairment in this enzyme’s activity leads to negative changes in TG, LDL, HDL, and VLDL levels and an increased saturated to unsaturated fatty acid ratio [152]. The hypercholesterolemic effect of insufficient Mg status may also be explained by regulation of the gene expression of LDLR (LDL receptor) and other transcription factors, such as sterol regulatory element-binding protein (SREBP)-1a and SREBP-2. Increased LDLR and SREBP expression at least mediate elevated LDL concentration [162,163]. The study among T1DM showed that Mg level was associated with the serum-oxidized low-density lipoprotein (ox-LDL) concentrations [164].

Most evidence regarding dependencies between Mg and the serum lipids profile comes from clinical studies on oral Mg supplementation. The results of these studies are not always consistent, which was emphasized by the authors of several meta-analyses focusing on describing this relationship (Table 4). The systematic review and meta-analysis by Asbaghi et al. [102] showed that Mg supplementation caused a significant reduction in serum LDL levels with no effect on TG, TC, and HDL, but subgroup analysis indicated that the effect of Mg treatment might be dose- and duration-dependent.

Difficulties in assessing the impact of Mg supply on the improvement of lipid metabolism may be caused by the high heterogeneity of the analyzed subjects in study groups. The meta-analysis of RCTs concerning the effect of oral Mg administration on glycemic control in patients with T2D showed that Mg supplementation increased HDL levels but had no effect on TC, LDL, and TG [104]. In another meta-analysis, no significant effects on plasma concentrations of TC, LDL, HDL, and TG concentrations of either diabetic or non-diabetic individuals were observed. However, when the analysis was stratified to compare subgroups of studies based on the presence of dyslipidemia, the significant positive effect of Mg treatment on LDL and TG level was observed in hypercholesterolemic (vs. non-hypercholesterolemic) and hypertriglyceridemic (vs. non-hypertriglyceridemic) patients [103] (Table 4). A recent review by Găman et al. [165] also pointed out a lack of consensus on whether a higher dietary intake of Mg or Mg supplementation affects the lipid profile and exerts lipid-lowering action. This extensive systematic review (a hundred and twenty-four studies) analyzing the association of Mg with lipids profile in patients with dyslipidemia and MetS-related disorders showed that in the majority of the studies conducted so far, there was no significant effect of Mg supplementation on HDL, LDL, TG, TC, TC/HDL-C, VLDL, or LDL-C/HDL-C. However, the authors emphasize that the interventions considered in the analysis were extremely heterogeneous, which made the inference much more difficult.

Several previous studies have focused on evaluating the link between serum Mg level, dietary Mg intake or Mg supplementation, and the development of lipid metabolism disorders. However, results obtained in this field are still ambiguous, mainly due to the significant heterogeneity of the studied groups, different methods of determining Mg concentration in the blood serum, or the lack of stratification of the compared patient subgroups.

## 8. Magnesium and Inflammation

Low-grade inflammation is defined as a C-reactive protein measurement above 3 mg/L, but below 10 mg/L [166]. Many previous studies have reported the importance of LGI as an underlying factor in the pathological processes of many metabolic disorders [167]. On the other hand, it is worth noting that inflammation as a part of innate immunity is a physiological response to cellular injury that is characterized by increased blood flow, capillary dilatation, leukocyte infiltration, and the expression of cytokines and other anti-inflammatory mediators, which lead to the repair of damaged tissue [168,169]. Many biochemical markers, e.g., the above-mentioned hs-CRP, white blood cell count (WBC), fibrinogen, IL-6, and TNF-α, are well-described biomarkers of low-grade or chronic inflammation. Several previous studies have shown that their concentration may depend on the Mg status. Hypomagnesemia is linked to many different physiopathological conditions in which LGI plays an important role in initiating and exacerbating the course of the disease [170], e.g., gastrointestinal disorders [171] and intestinal inflammation disease [172,173], obesity [174,175], or MetS [83,176]. Available studies provide data on an inverse relationship between serum Mg concentration and inflammatory markers [175,177].

The results of the 1999–2002 NHANES concerning slightly over 5000 children showed that those with an insufficient dietary Mg intake (less than 75% the RDA) was predisposed to increased serum CRP when compared to individuals with normal or high Mg intake [72]. Moreover, in another cross-sectional study including Mexican children, the low Mg serum was associated with elevated serum hs-CRP level [178]. Similar associations have been noticed in adult subjects in the 1999–2002 NHANES, especially in subgroups of overweight/obese subjects, in age > 40 years, and those consuming Mg at less than 50% of the RDA. These respondents were 2.24 times more likely to have elevated CRP than adults with a proper Mg intake [75]. In another cross-sectional study, it was noticed that the prevalence of hs-CRP ≥ 3 mg/L were 3–4 times more likely in the lowest tertile of Mg intakes [74]. Moreover, the systematic review and meta-analysis of cross-sectional studies by Dibaba DT. et al. [82] indicated that dietary Mg was significantly and inversely associated with serum CRP concentrations. In addition, a large body of further evidence confirmed that Mg intake was inversely associated with many different inflammation markers [71,73,172,178,179,180] (Table 3).

Data from in vivo and animal model studies allows us to describe the two major mechanisms by which Mg influences inflammation. As mentioned above, the deficit of Mg^2+^ leads to the activation of phagocytic cells, disturbances in the calcium channel-blocking effect, and increased cellular Ca concentration, activation of the NMDA receptor, and activation of the cellular inflammatory response [181]. This results in the secretion of many pro-inflammatory factors such as TNF-α, IL-1, IL-6, cytokine responders E-selectin, intracellular adhesion molecule-1, and vascular cell adhesion molecule-1 and stimulates the production of acute-phase proteins, e.g., CRP, fibrinogen [182]. Moreover, hypomagnesemia causes a systemic stress response through neuroendocrinological pathways (change or activation of acetylcholine, catecholamines, release of substance P (SP), and activation of renin-angiotensin system RAAS) [170,176]. The inflammation affects pro-atherogenic changes in the metabolism of lipoproteins (increase lipoproteins oxidation), endothelial dysfunction (increased production of peroxynitrite, which damages cellular biomolecules and structures) [183,184], and influences host metabolism, e.g., lipid peroxidation [170,185]. The second mechanism is associated with increased reactive oxygen species, promoting membrane oxidation and the nuclear factor kappa-light-chain-enhancer of activated B cells (NF-kB) production. Previous animal and clinical studies widely described that Mg deficiency is related to oxygen species stress markers [186,187,188]. Inappropriate expression of the NF-κB gene affects a wide spectrum of metabolic disturbances [170]. The relationship between Mg status and lipid metabolism and LGI is also because Mg is involved in regulating the gene expression of peroxisome proliferator-activated receptor gamma (PPAR-*γ*) [162]. PPARγ is a nuclear receptor regulating the transcription of several genes associated with fatty acid, carbohydrate, lipid, amino acids, and energy metabolism [189,190]. Additionally, PPARγ plays a key role in cell differentiation and apoptosis as well as modulates cytokines production by inhibiting the expression of pro-inflammatory molecules and the activities of other transcription factors, e.g., activator protein-1 (AP-1) and nuclear factor (NF)-kB [191]. The role of PPAR γ in endothelial dysfunction protection and the inflammatory response has also been described [192] (Figure 2).

Many previous studies have explained the effect of Mg supplementation on inflammatory markers, but their results are controversial. The RCTs among apparently healthy subjects with prediabetes and hypomagnesemia confirmed that low serum Mg correlated with higher CRP levels and showed that Mg supplementation led to an improvement in inflammatory status. After intervention, subjects from the supplemented group had higher serum Mg and lower serum hs-CRP than participants in the control group [193]. Positive changes after Mg administration have also been described by Nielsen et al. [179]. On the contrary, a systematic review and meta-analysis of RCTs by Simental-Mendia et al. [108] described that Mg treatment has no significant effect on plasma CRP concentrations; however, when the analysis was stratified to compare subgroups of studies in populations with LGI or without inflammation, a significant reduction in CRP level was observed in the first subgroup in comparison to the second group. Similar beneficial effects of oral Mg supplementation on CRP levels have been suggested by Mazidi et al. [107] (Table 4). The analysis noticed that changes in serum CRP levels were independent of the dosage and duration of Mg supplementation.

Conversely, the comprehensive systematic review and dose-response meta-analysis by Talebi et al. [105] did not confirm these results and described that oral Mg had no statistically significant effect on serum CRP, IL-6, and TNF-α when compared with controls. There was no association between the dose and duration of Mg administration and supplementation, and inflammation markers’ concentration (Table 4).

In summation, previous studies elucidated that low Mg plays a key role in the promotion of LGI. Additionally, data from several studies provided promising results on the positive effect of Mg intake on inflammation-related markers. However, there is a need to better investigate the relation between Mg therapy and inflammation, to determine whether routine supplementation should be prescribed to all patients with hypomagnesemia.

## 9. Conclusions

Mg is an essential nutrient for maintaining vital physiological functions. It is involved in many fundamental processes and Mg deficiency is often correlated with negative health outcomes. Hypomagnesemia may be implicated in the pathogenesis of metabolic disorders such as obesity, hypertension, diabetes, dyslipidemia, and low-grade inflammation. The majority of clinical studies confirmed the beneficial effect of Mg intake with diet and supplementation in the context of metabolic disorders. Although the exact mechanisms of Mg action remain to be established, targeting the Mg homeostasis may represent a new approach for the prevention and treatment of metabolic disorders and their complications. Nevertheless, more randomized controlled trials evaluating Mg supplementation strategies are needed.

## Figures and Tables

**Figure 1 nutrients-14-01714-f001:**
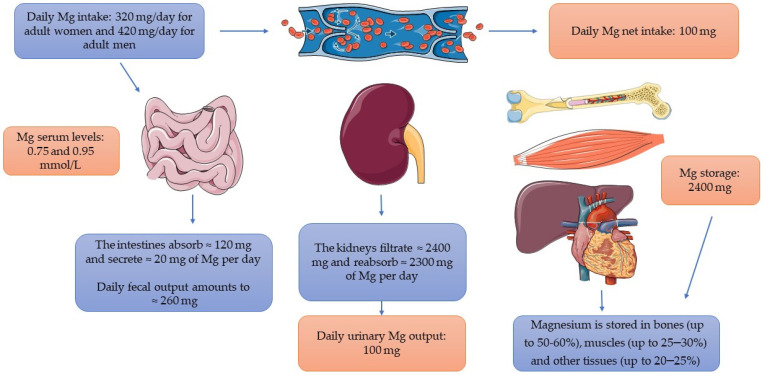
Magnesium turnover in the human body. Abbreviations: Mg, magnesium. This figure was made using the Servier Medical Art collection (http://smart.servier.com/) (accessed on 25 March 2022).

**Figure 2 nutrients-14-01714-f002:**
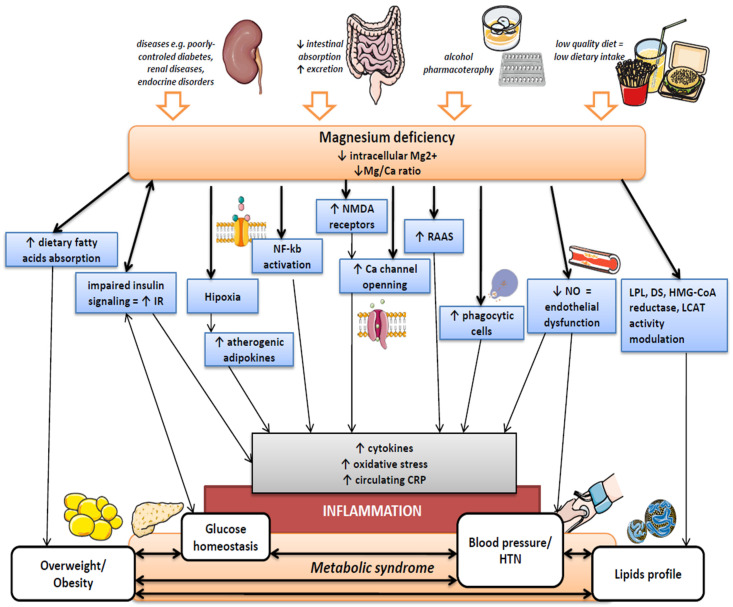
Influence of Mg deficiency on metabolic disorders. Abbreviations: Ca, calcium; CRP, C-reactive protein; DS, desaturase; HMG-CoA, β-hydroxy β-methylglutaryl-CoA; HTN, hypertension; IR, insulin resistance; LCAT, lecithin-cholesterol acyl transferase; LPL, lipoprotein lipase; Mg, magnesium; NF-kb, nuclear factor kappa-light-chain-enhancer of activated B cells; NMDA, N-methyl-D-aspartate; NO, nitric oxide; RAAS, renin-angiotensin system. This figure was made using the Servier Medical Art collection (http://smart.servier.com/) (accessed on 25 March 2022).

**Table 2 nutrients-14-01714-t002:** Conditions leading to hypomagnesemia.

Malabsorption	Crohn’s disease, ulcerative colitis, coeliac disease, short bowel syndrome, Whipple’s disease, chronic diarrhea, pancreatic insufficiency, inflammatory bowel diseases [2]
Endocrine disorders	Aldosteronism, hyperparathyroidism, hyperthyroidism, poorly-controlled diabetes [2,26]
Renal diseases	Chronic renal failure, dialysis, acute tubular necrosis, postobstructive diuresis, post kidney transplantation, excessive volume expansion, chronic metabolic acidosis [26,41]
Redistribution and intracellular shift	Refeeding syndrome, pregnancy, lactation, cardiopulmonary surgeries [41]
Medication use	Loop diuretics, aminoglycosides, amphotericin B, cyclosporine and tacrolimus, cisplatin, cetuximab, omeprazole, pentamidine [7,26]
Other causes	Inappropriate diet, chronic alcoholism, stress, severe burns [7,26]

**Table 3 nutrients-14-01714-t003:** Dietary magnesium intake and selected metabolic outcomes.

Metabolic Disorder	Reference-Year	Study Type	Population	Effects
Obesity	Guerrero-Romero, F. et al., 2022 [51]	Cross-sectional study	Metabolically healthy obese (MHO) individuals n-124 Metabolically unhealthy obese (MUO) n-123	The logistic regression analysis adjusted by sex and age showed that Mg intake is significantly associated with the MHO phenotype (OR = 1.17; 95% CI 1.07 to 1.25, *p* = 0.005)
	Naseeb, M. et al., 2021 [52]	Randomized, cluster-design study The HEALTHY Study	Ethnically diverse students (10–14 years) n-2181	Mg intake was related to BMI percentile at baseline and at end of the study (β = −0.05, 95% CI = −0.02 to 0, *p* = 0.04; β = −0.06, 95% CI = −0.02 to −0.003, *p* = 0.004); *R*^2^ (regression coefficient effect size) = 0.03; *R* ^2^ = 0.06) Mg intake was not related to plasma insulin and glucose concentrations
	Jiang, S. et al., 2020 [49]	Cross-sectional study National Health and Nutrition Examination Survey (NHANES) 2007–2014	Adult individuals (≥20 years) n-19,952	Mg intakes were negatively correlated with BMI (*p* < 0.05 at the quantiles of 0.1–0.9) and WC (*p* < 0.05 at 0.1–0.9 quantiles) after adjusting for age and gender
	Lu, L. et al., 2020 [50]	Multicenter longitudinal cohort study (30-year follow-up) The Coronary Artery Risk Development in Young Adults (CARDIA)	American young adults (18–30 years) n-5115	Compared to the lowest quintile (Q1) of Mg intake level, the incidence of obesity was reduced by 51% among participants in the highest quintile (Q5) [HR = 0.49, 95% CI = (0.40, 0.60), *p* for trend < 0.01]
	Castellanos-Gutiérrez, A. et al., 2018 [53]	Population-based multistage probabilistic study Mexican National Health and Nutrition Survey 2012	Adult individuals (20–65 years) n-1573	Increase in 10 mg per 1000 kcal/day of Mg was associated with an average decrease in BMI of 0.72% (95% CI: −1.36, − 0.08) and 0.49 cm (95% CI: −0.92, − 0.07) of WC An increase in Mg intake was associated with an average decrease in serum glucose by 0.59% (95% CI: −1.08, − 0.09)
Hypertension	Dominguez, L.J. et al., 2020 [54]	Prospective study The SUN Project (Seguimiento Universidad de Navarra)	Mediterranean population n-14,057	Dietary Mg intake < 200 mg/day was independently associated with a higher risk of developing high blood pressure, especially in overweight/obese participants
	Choi, M.-K. et al., 2015 [55]	Cross-sectional studyKorean National Health and Nutritional Examination Survey data (2007–2009)	Adults participants (20 years and over) n-11,685	No significant association between dietary Mg intake and the risk of HTN in obese women after adjusting relevant factors, the adjusted odds ratio of DBP prevalence in the highest magnesium intake quartile was 0.40 compared with the lowest magnesium intake quartile (95% CI = 0.25–0.63, *p* for trend = 0.0014)
	Huitrón-Bravo, G.G. et al., 2015 [56]	Cohort study Health Workers Cohort Study	Mexican adult subjects n-1378	Trend of decreasing DBP with rising Mg intake, by tertiles (the coefficients were −0.75 mmHg [95% confidence interval (CI): −1.77, 0.27], −1.27 mmHg (95% CI: −2.73, −0.02; *p* for trend = 0.01) was found In the fully adjusted model, Mg intake was inversely associated, although not significantly, with the risk of developing hypertension; subjects in the highest tertile of Mg intake had a decreased risk for HTN (odds ratio 0.83, 95% CI: 0.49–1.39, *p* for trend = 0.48)
	Chacko, S. A. et al., 2010 [57]	Cross-sectional study Women’s Health Initiative Observational Study	Postmenopausal women (50–79 years) n-3713	An inverse relationship between Mg intake and plasma concentrations of soluble vascular cell adhesion molecule 1 (sVCAM-1) and E-selectin An increase of 100 mg Mg per day was inversely associated with sVCAM-1 (−0.04 ± 0.02 ng/mL; *p* = 0.07) and other markers of inflammation as hs-CRP, IL-6 or TNF-α
Type 2 diabetes	Huang, W. et al., 2021 [58]	Cross-sectional study National Health and Nutrition Examination Survey-NHANES (2007–2014)	Adults participants n-10,249	The association of serum vitamin D with the incidence of T2D appeared to differ between the low Mg intake group and the high Mg intake group (OR: 0.968, 95%Cl: 0.919–1.02 vs. OR: 0.925, 95%Cl: 0.883–0.97) There was evidence of interaction between vitamin D levels and Mg intake on decreasing the incidence of T2D (*p*-value for interaction = 0.001)
	Gant, C. M. et al., 2018 [59]	Cross-sectional study DIAbetes and LifEstyle Cohort Twente-1	T2DM patients n-450 (mean age 63 ± 9 years) diabetes duration of 11 (7–18) years)	Adjusted coronary heart disease (CHD) prevalence ratios for the highest compared to the lowest quartiles were 0.40 (0.20, 0.79) for Mg intake, 0.63 (0.32, 1.26) for 24 h urinary Mg excretion, and 0.62 (0.32, 1.20) for plasma Mg concentration For every 10 mg increase of Mg intake from vegetables, the prevalence of CHD was, statistically non-significantly, lower [0.75 (0.52, 1.08)]
	Hruby, A. et al., 2017 [60]	Prospective cohort study Results From Three U.S. Cohorts: The Nurses’ Health Study (1984–2012), NHS2 (1991–2013) and the Health Professionals’ Follow-Up Study (1986–2012)	Incident cases of T2DM over 28 years of follow-up n-17,130	In pooled analyses across the three cohorts, those with the highest magnesium intake had 15% lower risk of type 2 diabetes compared with those with the lowest intake (pooled multivariate HR in quintile 5 vs. 1: 0.85 [95% CI 0.80–0.91], *p* < 0.0001).
	Konishi, K. et al., 2017 [61]	Population-based cohort study	Adults participants 13,525	Compared with women in the low quartile of Mg intake, women in the high quartile were at a significantly reduced risk of diabetes (HR 0.50; 95% CI 0.30–0.84; *p*-trend 0.005) after adjustments for covariates. In men, there was no association between magnesium intake and the risk of diabetes
	Huang, J.H. et al., 2012 [62]	Cross-sectional study	T2DM patients (65 years and above) n-210	Among type 2 diabetes patients (n-201), 88.6% had Mg intake under the RDA, while 37.1% had measurable hypomagnesemia. Moreover, Mg intake was associated with metabolic syndrome components, i.a., positively with high-density lipoprotein level (HDL, *p* = 0.005) and negatively with triglycerides (TG), WC, body fat percent and BMI (*p* < 0.005)
	Kim, D.J. et al., 2010 [63]	Prospective cohort study	Adult Americans (18–30 years) n-4497	The multivariable-adjusted hazard ratio of diabetes for participants in the highest quintile of Mg intake was 0.53 (95% CI, 0.32–0.86; *p*-trend < 0.01) compared with those in the lowest quintile Mg intake was significantly inversely associated with hs-CRP, IL-6, fibrinogen, and HOMA-IR, and serum magnesium levels were inversely correlated with hs-CRP and HOMA-IR
Dyslipidemia	Jin, H. et al., 2018 [64]	Retrospective National Health and Nutrition Examination Study (NHANES) 2001–2013	Adult individuals n-12,284	Dietary Mg was positively correlated with HDL concentration but negatively with the TC/HDL-C ratio in females
	Bain. L. et al., 2015 [65]	cross-sectional study (European Prospective Investigation into Cancer (EPIC)-Norfolk)	Adult individuals (40–75 years) n-4443	Inverse relationship between high dietary Mg intake (mean 456 mg/d) and serum TC (*p*-trend = 0.02 men and 0.04 women)
	Yamori, Y. et al., 2015 [66]	Cross-sectional analysis (World Health Organization-coordinated Cardiovascular Diseases and Alimentary Comparison (CARDIAC) Study (1985–1994)	Adult participants (48–56 years) n-4211	Mg/creatinine (Cre) ratio was inversely associated with BMI, SBP, DBP, and TC (*p* for linear trend < 0.001 for each)
	Mirmiran, P. et al., 2012 [67]	Cross-sectional study Tehran Lipid and Glucose Study 2006–2008	Tehran healthy adults (18–74 years) n-2504	Higher dietary Mg was inversely correlated with TG level (*p* = 0.009), but there was no association with HDL (*p* = 0.44) Dietary Mg intake was inversely associated with FBG (*p* = 0.006) and WC (0.006)
	Ohira, T. et al., 2009 [68]	Cross-sectional study Atherosclerosis Risk in Communities Study cohort	Adult participants (45–64 years) n-14,221	Higher Dietary Mg intake was inversely associated with LDL (*p* = 0.01) and positive correlated with HDL (*p* = 0.001)
	Ma, J. et al., 1995 [69]	Cross-sectional study (Atherosclerosis Risk in Communities (ARIC) Study)	Adult participants (45–64 years) n-15,248	Dietary Mg intake was inversely associated with fasting serum insulin, HDL, SBP, DBP
	Singh, R.B. et al., 1990 [70]	RCTs	High-risk of CDV adults n-430	Positive changes in TC, LDL, and TG (10.1%) and slight elevation in HDL in the Mg-rich diet group
Low-grade inflammation	Arablou, T. et al., 2019 [71]	Cross-sectional study	Patients with active rheumatoid arthritis n-87	Significant negative correlations were observed between Mg intake with PGE2 (R = −0.24)], IL-1β (R = −0.23), and IL-2 (R = −0.25)
	King, D.E. et al., 2007 [72]	Cross-sectional, nationally representative National Health and Nutrition Examination Survey (NHANES 1999–2002)	Children (6–17 years) n-5007	Insufficient dietary Mg intake (less than 75% the RDA) was associated with higher CRP (OR: 1.58, 95% CI: 1.07-infinity, *p* < 0.005)
	Song, Y. et al., 2007 [73]	Cross-sectional study(Nurses’ Health Study)	Women (43–69 years) n-657	Mg intake was inversely associated with: CRP (*p* = 0.003), E-selectin (*p* = 0.001), and sICAM-1 (*p* = 0.03) Women in the highest quintile of dietary Mg intake were 24% lower for CRP (*p* = 0.03) and 14% lower for E-selectin (*p* for trend = 0.01) than those for women in the lowest quintile
	Bo, S. et al., 2006 [74]	Cross-sectional study	Adult subjects n-1653	Prevalence of hs-CRP ≥ 3 mg/L were 3–4 times as likely in the lowest tertile of magnesium intakes
	King, D.E. et al., 2005 [75]	Cross-sectional study (NHANES 1999–2002)	Adult subjects ≥ 17 years) n-5773	Insufficient dietary Mg intake (less than 50% of the RDA) was associated with higher CRP (95% CI: 1.13, 4.46)

Abbreviations: BMI, body mass index; CHD, coronary heart disease; Cre, creatinine; CRP, C-reactive protein; DBP, diastolic blood pressure; FBG, fasting blood glucose; HDL, high-density lipoprotein; HOMA-IR, Homeostatic Model Assessment for Insulin Resistance; HR, hazard ratio; HTN, hypertension; IL-1, interleukin 1; IL-6, interleukin 6; IR, insulin resistance; LDL, low-density lipoprotein; Mg, magnesium; MHO, metabolically healthy obese; MUO, metabolically unhealthy obese; PGE2, prostaglandin E2; RDA, recommended daily allowance; SBP; systolic blood pressure; sICAM-1, soluble intercellular adhesion molecule 1; sVCAM-1, soluble vascular cell adhesion molecule 1; T2DM, type 2 diabetes mellitus; TC, total cholesterol; TG, triglycerides; TNF-α, tumor necrosis factor-α; WC, waist circumference.

**Table 4 nutrients-14-01714-t004:** Magnesium supplementation in management of MetS components—the reviews of meta-analysis.

Metabolic Disorder	Reference-Year	Study Type	Population	Effects
Obesity	Askari, M. et al., 2021 [91]	32 RCTs	Adult participants n-2551 Doses: 48–450 mg/d Duration: 6–24 weeks	Mg supplementation resulted in a slight reduction in BMI (WMD: −0.21 kg/m^2^, 95% CI: −0.41, −0.001, *p* = 0.048)
	Rafiee, M. et al., 2021 [92]	28 RCTs	Adult participants n-2013	No significant changes in anthropometric indicators after Mg supplementation in the overall analysis In the subgroup Mg supplementation decreases WC in obese individuals (twelve trials, n-997, WMD = −2.09 cm, 95% CI: –4.12, −0.07, *p* = 0.040; I2 = 0%)
	Asbaghi, O. et al., 2021 [93]	11 RCTs	Patients with T2DM n-673	Mg supplementation did not significantly change body weight (WMD: −0.01 kg, 95% CI: −0.36 to 0.33), BMI (WMD: −0.07, 95% CI: −0.18 to 0.04) or WC (WMD: 0.12, 95% CI: −1.24 to 1.48) Mg supplementation reduced the SBP (WMD: −5.78 mmHg, 95% CI: −11.37, −0.19) and DBP (WMD: −2.50 mmHg, 95% CI: −4.58, −0.41) in T2DM patients
Hypertension	Dibaba, D.T., 2017 [95]	11 RCTs	Individuals with insulin resistance, prediabetes or NCDs n-543 Median duration: 3.6 months Doses: 365–450 mg/day	Mg supplementation resulted in a mean reduction in SBP of 4.18 mmHg (standardized mean differences, SMD: −0.20; 95% CI: −0.37, −0.03) and 2.27 mmHg in DBP (SMD: −0.29; 95% CI: −0.46, −0.12)
	Zhnag, X. et al., 2016 [96]	34 RCTs	Normotensive and hypertensive adults n-2028 Duration: 3 months Doses: 368 mg/d	Mg supplementation significantly reduces SBP by 2.00 mmHg (95% CI: 0.43, 3.58) and DBP by 1.78 mmHg (95% CI 0.73, 2.82) along with increase in serum Mg concentration by 0.05 mmol/L (95% CI: 0.03, 0.07)
	Kass, L. et al., 2012 [97]	22 RCTs	Adults participants n-1173 Duration: 3 to 24 weeks of follow-up Mean doses: 410 mg/d	A small reduction in SBP (0.32, 95% CI: 0.23, 0.41) and DBP (0.36, 95% CI: 0.27, 0.44) with a greater effect for the intervention in crossover trials (DBP 0.47, SBP 0.51)
Diabetes	Veronese, N. et al., 2021 [98]	25 RCTs	Diabetic participants (13 studies) n-361 Mg treatment n-359 placebo Duration: median of 12 weeks (range: 4–48) Participants at high risk of developing diabetes (12 studies) n-477 Mg treatment n-480 placebo Duration: median of 14 (range: 4–24) weeks	Treatment with Mg significantly reduced FBG in 325 participants with diabetes compared to 331 taking placebo (n = 11 studies; SMD = −0.426; 95%CI: −0.782 to −0.07; *p* = 0.02), this finding was characterized by a high heterogeneity (I2 = 79.0%) Mg supplementation did not improve HbA1c in 301 participants compared to 307 participants taking placebo (n = 10 studies; SMD = −0.134; 95%CI: −0.409 to 0.141; *p* = 0.34; I2 = 63.7%) Mg supplementation significantly improved FBG in 482 subjects at high risk of diabetes compared to 485 randomized to placebo (11 RCTs; SMD = −0.344; 95%CI: −0.655 to −0.03; *p* < 0.0001; I2 = 81.2%) Similarly, Mg significantly improved 2h OGTT in 3 studies involving 210 participants (SMD = −0.35; 95%CI: −0.62 to −0.07; I2 = 0%) Mg significantly decreases HOMA-IR in 9 studies (340 Mg vs. 344 placebo) (SMD = −0.234; 95%CI: −0.443 to −0.025; *p* = 0.028; I2 = 43.2%)
	Verma, H. et al., 2017 [99]	24 RCTs	Diabetic and non-diabetic individuals n-1694	Significant improvement in: FBG (WMD) = −4.641 mg dL^−1^, 95% confidence interval (CI) = −7.602, −1.680, *p* = 0.002), HDL (WMD = 3.197 mg dL^−1^, 95% CI = 1.455, 4.938, *p* < 0.001), LDL (WMD = −10.668 mg dL^−1^, 95% CI = −19.108, −2.228, *p* = 0.013), TG(WMD = −15.323 mg dL^−1^, 95% CI = −28.821, −1.826, *p* = 0.026) SBP (WMD = −3.056 mmHg, 95% CI = −5.509, −0.603, *p* = 0.015). During subgroup analysis, a more beneficial effect of magnesium supplementation was observed in diabetic subjects with hypomagnesaemia
	Veronese, N. et al., 2016 [100]	18 RCTs	Individuals with T2DM n-336 Mg treatment n-334 placebo People at high risk of diabetes n-226 Mg treatment n-227 placebo	Mg supplementation influence beneficial on glucose parameters in people with T2DM: reduced FBG (SMD: −0.40; 95% CI: −0.80, −0.00; I2 = 77%-9 studies) Mg supplementation influence beneficial on insulin sensitivity parameters in people at high risk of diabetes: improved FBG after a 2 h oral glucose tolerance test (SMD: −0.35; 95% CI: −0.62, −0.07; I2 = 0%-3 studies) and reductions in HOMA-IR (SMD: −0.57; 95% CI: −1.17, 0.03; I2 = 88%-5 studies)
Dyslipidemia	Tan, X et al., 2022 [101]	4 RCTs	Gestational diabetes n-ns Duration: 4–26 weeks Doses: 250–500 mg/d	Mg supplementation significantly reduced: FBG (SMD) = −0.99; 95% confidence interval (CI) = −1.28 to −0.70; *p* < 0.00001), serum insulin (SMD = −0.75; 95% CI = −1.24 to −0.26; *p* = 0.003), HOMA-IR (SMD = −0.74; 95% CI = −1.10 to −0.39; *p* < 0.0001) LDL (SMD = −0.39; 95% CI = −0.73 to −0.04; *p* = 0.03) TC (SMD = −0.62; 95% CI = −0.97 to −0.27; *p* = 0.0005) and increased quantitative insulin sensitivity check index (SMD = 0.47; 95% CI = 0.12 to 0.82; *p* = 0.008).
	Asbaghi, O. et al., 2021 [102]	12 RCTs	Patients with T2DM n-677	Significant reduction in serum LDL levels (*p* = 0.006) with no effect on TG, TC, and LDL Effect of Mg supplementation might be dose- and duration-dependent: Mg supplementation lasting >12 weeks led to decreasing the serum TC (*p* = 0.002), Mg in dose <300 mg/day significantly decreased the serum LDL (*p* < 0.001), Mg in dose >300 mg/day markedly increased the serum HDL levels (*p* = 0.026) Inorganic Mg supplementation (vs. organic Mg) led to improvement in LDL (*p* < 0.001) and TC (*p* = 0.003)
	Simental-Mendía, L.E. et al., 2017 [103]	18 RCTs	Diabetic and non-diabetic adults n-1192 Duration: 2–6 months Doses: 300–730 mg/d	No significant effect on: TC (WMD 0.03 mmol/L, 95% CI −0.11, 0.16, *p* = 0.671), LDL (WMD −0.01 mmol/L, 95% CI −0.13, 0.11, *p* = 0.903), HDL (WMD 0.03 mmol/L, 95% CI −0.003, 0.06, *p* = 0.076), TG (WMD −0.10 mmol/L, 95% CI −0.25, 0.04, *p* = 0.149).
	Song, Y. et al., 2006 [104]	9 RCTs	Patients with T2DM n-370 Duration: 4–16 weeks Median doses: 360 mg/day	Mg supplementation increased HDL levels (WMD: 0.08 mmol/L (95% CI: 0.03, 0.14); *p* for heterogeneity = 0.36) but had no effect on TC, LDL and TG Lower FBG (WMD: −0.56 mmol/l (95% CI, −1.10 to −0.01); *p* for heterogeneity = 0.02) with no effect on HbA1c [−0.31% (95% CI, −0.81 to 0.19); *p* for heterogeneity = 0.10], SBP, DBP.
Low-grade inflammation	Talebi, S. et al., 2022 [105]	18 RCTs	Adult participants n-927 Duration: 4–26 weeks Doses: 20–500 mg/d	Mg supplementation had no statistically significant effect on serum CRP (WMD,−0.49; 95% CI,−1.72 to 0.75 mg/L; *p* = 0.44), IL-6 (WMD,−0.03; 95% CI,−0.40 to 0.33 pg/mL; *p* = 0.86), and TNF-α (WMD, 0.12; 95% CI,−0.08 to 0.31 pg/mL; *p* = 0.24).
	Veronese, N. et al., 2022 [106]	17 RCTs	Adult participants n-889	Mg supplementation significantly decreased serum CRP (SMD = −0.356; 95% CI: −0.659 to −0.054; *p* = 0.02), and increased nitric oxide (NO) levels (SMD = 0.321; 95% CI: 0.037 to 0.604; *p* = 0.026)
	Mazidi, M. et al., 2018 [107]	8 RCTs	Adult participants n-349 Duration: 8 h to 6.5 months Doses: 320–1500 mg/d	Mg supplementation led to reduction in CRP (WMD: −1.33 mg/L; 95% CI: −2.63, −0.02, heterogeneity *p* < 0.123; I2 = 29.1%) Changes in serum CRP levels were independent of the dosage and duration of Mg supplementation
	Simental-Mendía, L.E. et al., 2017 [108]	11 RCTs	Adults participants n-ns	Mg treatment was found to significantly affect plasma concentrations of CRP in subgroups of populations with baseline plasma CRP > 3 mg/L (WMD: −1.12 mg/L, 95% CI: −2.05, −0.18, *p* = 0.019)

Abbreviation: BMI, body mass index; CRP, C-reactive protein; DBP, diastolic blood pressure; FBG, fasting blood glucose; HDL, high-density lipoprotein; HOMA-IR, Homeostatic Model Assessment for Insulin Resistance; HbA1c, hemoglobin A1c; HTN, hypertension; IL-1, interleukin 1; IL-6, interleukin 6; IR, insulin resistance; LDL, low-density lipoprotein; Mg, magnesium; NO, nitric oxide; OGTT, ns, not found; oral glucose tolerance test; RCTs, randomized controlled trials; RDA, recommended daily allowance; SBP; systolic blood pressure; SMD, standardized mean differences; T2DM, type 2 diabetes mellitus; TC, total cholesterol; TG, triglycerides; TNF-α, tumor necrosis factor-α; WC, waist circumference; WMD, weighted mean differences.

## Data Availability

Not applicable.

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
