# Peer review of "The Role of Magnesium in the Pathogenesis of Metabolic Disorders"

_nutrients, 2022, doi:10.3390/nu14091714_

Round 1

Reviewer 1 Report

I congratulate the authors for the huge work done.

Here are present my suggestion and comment:

  1. Abstract, line 12, not are only some studies, but there are many studies, so use a more incisive sentence.
  2. Abstract, line16 the protective action of Magnesium (Mg) does not include limiting the adipose tissue enlargement, it is not discussed in your work.
  3. line 28 and subsequent lines. Mg is not a macronutrient, but it is a micronutrient, in particular macroelement. Please change the term macronutrient to micronutrient in the whole text.
  4. line 30 you change the word 'transformation' with 'metabolism', it is more correct.
  5. line 33, RDA varies from countries to countries read this paper (Cazzola R, Della Porta M, Manoni M, Iotti S, Pinotti L, Maier JA. Going to the roots of reduced magnesium dietary intake: A tradeoff between climate changes and sources. Heliyon. 2020;6(11):e05390. Published 2020 Nov 3. doi:10.1016/j.heliyon.2020.e05390) , rephrase the sentence adding the other RDA (i.e. Europen).
  6. line 41, ... spectrum of diseases. adding the examples, CVD, Depression, Migraine, etc...
  7. Line 48, add a more recent reference to support the sentence. e.g. Piuri G, Zocchi M, Della Porta M, Ficara V, Manoni M, Zuccotti GV, Pinotti L, Maier JA, Cazzola R. Magnesium in Obesity, Metabolic Syndrome, and Type 2 Diabetes. Nutrients. 2021 Jan 22;13(2):320. doi: 10.3390/nu13020320. PMID: 33499378; PMCID: PMC7912442.
  8. line 59, insert the amount in the blood (less than 1%) ( read the papers previously cited).
  9. line 61 the serum reference ranges are updated to 0.75 to 0.95 mmol/L read this paper DiNicolantonio JJ, O’Keefe JH, Wilson W. Subclinical magnesium deficiency: a principal driver of cardiovascular disease and a public health crisis. Open Heart 2018;5:e000668. doi:10.1136/ openhrt-2017-000668
  10. line 67 you substitute chronic Mg deficiency with subclinical Mg deficiency, it is more correct.
  11. line 84 cardigan fluctuation may be a typo, you intend circadian fluctuation?
  12. line 103 the Mg deficiency does not result only from inappropriate nutrition but even from soil pauperization, please read this paper (Cazzola R, Della Porta M, Manoni M, Iotti S, Pinotti L, Maier JA. Going to the roots of reduced magnesium dietary intake: A tradeoff between climate changes and sources. Heliyon. 2020;6(11):e05390. Published 2020 Nov 3. doi:10.1016/j.heliyon.2020.e05390).
  13. Line 142 The reference Gruber U. et al 2015 is not written in the same manner as the others [n°].
  14. line 146 Better explain the role of Vitamin D and PTH in Mg homeostasis.
  15. line 171 please insert a high intake of zinc in the list of the decreasing Mg absorption factors, as reported in ref 25.
  16. Table 1 a) Change reference 0.7 to 0.75 and 1.00 to 0.95 mmol/L.

                 b) Insert the references in the legenda of table 1

                 c) In the source of Mg add pulses.

  17. line 184, High content of adipose tissue measured by? Insert the method to measure adipose tissue, DXA? BIA? skinfold caliper?
  18. lines 207-209, this sentence is not supported by ref 45. rephrase the sentence. e.g in overweight/obese children, the Mg deficiency depends on decreased absorption etc...
  19. lines 211-213 insert a reference that experimentally supports the sentences or eliminate these sentences. reference 41 only cites this hypothesis and does not test it.
  20. line 248 -0.21 kg/m2 is not a greater reduction, it is a slight reduction.
  21. line 475 insert the acronym HDL.
  22. Lines 538-543 In the ref 123 study has investigated the effect of the fibre? Do the 2 diets, the usual diet vs the Mg-rich diet, have the same amount of fibre? fibre has a greater effect on HDL and LDL and TG.
  23. lines 566-567. there is a mistake in the sentence the reduction is on LDL or HDL?
  24. line 685 grammatical error 'this results'.

Reviewer 2 Report

This review discussed the correlation between Mg deficiency and obesity, hypertension, diabetes, dyslipidemia, low grade inflammation and metabolic disorders. Plenty of literature study was performed.

Major:

  • Section “2. Magnesium status in the human body”: the information in this section is not really about Mg status. Instead, different methods to test Mg levels are discussed with contrary reports. This section may be divided to two sections. Also the discussion about the methods were only listing methods without clarifying which method is better used under which situation and why it is better than other methods.
  • Language needs to be improved. For example, line 102-103: two “especially”, needs editing; line 104, “also” should be removed; line 137, “what” should be “that”; line 156 “what is important” and line 513 “what’s more”, these are sentences, not adv, and cannot be used to start a sentence. Too many these types of issues.
  • Line 207: why overweight people have decreased absorption or increased excretion of Mg?
  • Line 211: does Mg deficiency cause obesity or obesity results in Mg deficiency? Which is the cause and which is the result?
  • A table of what clinical conditions/diseases could cause Mg deficiency may be helpful.
  • In each section, using tables to list and summarize literature studies can give readers better idea of how many studies support the correlation between Mg deficiency/supplementation and diseases, and how many studies showed no correlations. Then in the text, only explain one or two examples for each statement will be enough.

Minor:

  • Repeated sentences in Abstract and Introduction.
  • Line 29 “, i.e.,” should be removed
  • Line 32 and 48, above mentioned should be above-mentioned.
  • Line 62: “what is important it is estimated…..”, the sentence needs to be corrected.
  • Line 67, 500, 552, 685, “contrary” is adj, try “on the contrary”.
  • Mg has been abbreviated, but magnesium was still used here and there.
  • References should be given for the statement in line 74-76.
  • Line 131, should be “macronutrient” as in the whole manuscript.
  • There are many other typing errors or grammar issues as well.

Round 2

Reviewer 1 Report

The authors answered and corrected the entire manuscript, now this paper is ready to be published.